# Laser–Material Interactions of High-Quality Ultrashort Pulsed Vector Vortex Beams

**DOI:** 10.3390/mi12040376

**Published:** 2021-04-01

**Authors:** Yue Tang, Walter Perrie, David Rico Sierra, Qianliang Li, Dun Liu, Stuart P. Edwardson, Geoff Dearden

**Affiliations:** 1Laser Group, School of Engineering, University of Liverpool, Brownlow Street, Liverpool L69 3GQ, UK; hsytan10@liverpool.ac.uk (Y.T.); sgdricos@liverpool.ac.uk (D.R.S.); psqli7@liverpool.ac.uk (Q.L.); me0u5040@liverpool.ac.uk (S.P.E.); gdearden@liverpool.ac.uk (G.D.); 2Laser Group, School of Mechanical Engineering, Hubei University of Technology, Wuhan 430068, China; dun.liu@hbut.edu.cn

**Keywords:** multiple beam generation, spatial light modulator, cylindrical vector, concurrence, ablation threshold, LIPSS

## Abstract

Diffractive multi-beams based on 1 × 5 and 2 × 2 binary Dammann gratings applied to a spatial light modulator (SLM) combined with a nanostructured S-wave plate have been used to generate uniform multiple cylindrical vector beams with radial and azimuthal polarizations. The vector quality factor (concurrence) of the single vector vortex beam was found to be C = 0.95 ± 0.02, hence showing a high degree of vector purity. The multi-beams have been used to ablate polished metal samples (Ti-6Al-4V) with laser-induced periodic surface structures (LIPSS), which confirm the polarization states unambiguously. The measured ablation thresholds of the ring mode radial and azimuthal polarizations are close to those of a Gaussian mode when allowance is made for the expected absolute intensity distribution of a ring beam generated from a Gaussian. In addition, ring mode vortex beams with varying orbital angular momentum (OAM) exhibit the same ablation threshold on titanium alloy. Beam scanning with ring modes for surface LIPSS formation can increase micro-structuring throughput by optimizing fluence over a larger effective beam diameter. The comparison of each machined spot was analysed with a machine learning method—cosine similarity—which confirmed the degree of spatial uniformity achieved, reaching cos*θ* > 0.96 and 0.92 for the 1 × 5 and 2 × 2 arrays, respectively. Scanning electron microscopy (SEM), optical microscopy and white light surface profiling were used to characterize and quantify the effects of surface modification.

## 1. Introduction

The ability to create complex spatial and vector optical fields is accelerating the progress of scientific fields such as encoding information in optical communications [1], sub-diffraction high-resolution imaging [2], particle acceleration [3], improving stability of NL filamentation in dielectrics [4] and sophisticated control of light–matter interactions such as an optical spanner [5,6]. Such states of light, also able to carry high levels of orbital angular momentum (OAM), can be created externally from a Gaussian beam with uniform polarization by the use of spatial light modulators, spin–orbit components such as liquid crystal Q plates [7,8], nanostructured S-waveplates [9] and complex metasurfaces [10]. Very recently, however, a meta-surface (J plate) using intracavity in a compact laser has led to direct control of the angular momentum of visible light with OAM quantum numbers *l* >100 as well as non-symmetric vector vortex beams lasing simultaneously on independent OAM states with Δ*l* = 90 apart [11]. This remarkable development is set to continue [12].

While Gaussian beams are often linearly polarized, the use of cylindrical vector vortex (CVV) beams in ultrafast laser–material ablation can yield complex features through laser-induced periodic surface structures (LIPSS) [13], which are both polarization and fluence dependent. Inhomogeneous polarization states, such as radial and azimuthal polarization, which have a phase singularity and hence an intensity null on axis, have been used to improve laser cutting [14] and, more recently, in producing bio-mimetic surfaces with femtosecond pulses combined with Q plates. The use of single beam can severely limit processing speed; hence the creation of parallel beams can reduce fabrication time significantly whether using a fixed diffractive optical component (DOE) [15] or a programmable spatial light modulator (SLM), demonstrated first on surfaces [16,17] and in material inscription [18,19]. Cylindrical vector beams can be used to create more complex surface plasmonic structures than linearly polarized Gaussian beams [20]. For a given pulse energy, the ring modes reduce maximum fluence on target so that higher pulse energies can be used, increasing light usage and speeding up processing. Ring modes can also reduce the wall angle during laser drilling [14]. The laser cutting efficiency for a radially polarized beam was shown to be higher than a circularly polarized Gaussian beam [21]. A clearer outline of the ablated zone with less debris was found while using ring mode vortex ablation [22]. The extension to multi-VV beams is therefore important, but the beam quality of these cylindrical vector (CV) beams also needs to be measured.

While computer-generated holograms (CGHs) calculated by inverse Fourier transforms (IFTs) [23] generate a 0–2π phase map, multi-spot uniformity can still be an issue [24]. One of the most efficient ways to improve the uniformity of multi-spots can be realized by iterative design of sub-wavelength metasurfaces [25]. By contrast, binary-phase holograms, such as Dammann gratings, are much simpler and can produce equal-intensity spots at diffractive orders with good uniformity [26,27] due to their phase changes involving only 0 and π. In this paper, an SLM addressed with binary-phase computer-generated holograms, combined with a nanostructured geometric phase plate, has been used to generate single and multiple uniform 1 × 5 and 2 × 2 arrays with plane-wave radial and azimuthal polarizations. LIPSS patterns were imprinted on titanium (Ti-6Al-4V) samples, which confirmed the vector polarization states. A careful analysis of theoretically expected and experimentally measured ring and Gaussian mode intensity distributions allowed a comparison of ablation thresholds with scalar and vector ring beams, which indicate that the ablation threshold is mode independent for picosecond laser ablation on the titanium alloy Ti-6Al-4V.

## 2. Experimental

### 2.1. Experimental Setup

Figure 1 shows a schematic of the experimental setup. The ultrafast laser system used for this research is a custom-made Nd:VAN seeded regenerative amplifier laser (High-Q IC-355-800 ps, Photonic Solutions, Edinburgh, Scotland), with horizontal linear polarization: *τ* = 10 ps, *λ* = 1064 nm, PRF = 10 kHz, and *M*_0_^2^ = 1.3. The output is attenuated, then passes through a beam expander (*M* ≈ ×3), reflected from two plane mirrors (M2 and M3) for controlling the beam path and illuminated at low AOI on a reflective phase-only SLM, Hamamatsu X10468-03 (800 × 600 pixels and dielectric coated for 1064 nm, Hamamatsu, UK). A combined hologram with a desired Dammann grating and Fresnel lens was applied on the SLM, the latter used to defocus the diffracted beams relative to the central zero order to improve array uniformity. The beams were then passed through a nanostructured S-wave plate (SWP; RPC-1030-10-109, Altechna, Vilnius, Lithuania) to modulate the polarization state to radial or azimuthal by rotating the SWP fast axis. The SWP was carefully placed at the 4f image plane of lens L2, where the various diffracted Gaussian spots overlapped precisely, thus generating the required array of vector vortex beams. The SWP, a 25 mm fused silica disk, inscribed up to 10 mm in diameter, was mounted on a fine x,y translator for accurate centering. The optic axis was marked, thus allowing variation from full vector to scalar beams when rotating the optic. The distance between the SWP and SLM2 was minimized, so that the beams very nearly overlapped on SLM2. The angular offset between the diffracted beams with the 1 × 5 array was Δθ ~ 1 × 10^−3^ radian, so that with an SWP–SLM2 separation of 180 mm, the maximum offset of the outer spots at SLM2 was Δs = ±0.4 mm, 5% of the spot diameters at the SLM (8 mm).

A flip mirror was placed after the S-wave plate to direct the beams to a CCD-based beam profiler (SP620U, Spiricon, Jerusalem, Israel) with a focusing lens (L3, f = 500 mm) and polarizer (PL) used to analyse the polarization states at the Fourier plane. With the flip mirror removed, a 4f imaging system (L4 and L5, f = 400 mm) imaged the SLM complex field to the input aperture of a galvo scanner (Nutfield-XLR8-14, Hudson, NY, USA) via periscope mirrors M4 and M5 and focused with a flat field f-theta lens (f = 100 mm, Linos).

The samples were mounted on a precision 5-axis (x, y, z, u, v) motion control system (A3200 Ndrive system, Aerotech) for precise location of the focal plane. Polished Ti-6Al-4V and thin-film chromium on glass were used as sample substrates.

Calculation of the Dammann gratings was accomplished in MATLAB, with numerical data used from Zhou et al. [28]. With the nanostructured S-wave plate positioned close to the SLM and low angles of diffraction, multi-cylindrical vector beam arrays were generated.

### 2.2. Measurement of Single Vector Vortex Beam

While the beam quality factor for Gaussian and multi-mode beams is well understood, these beams are scalar with fixed polarizations. On the other hand, complex beams such as VV beams with varying spatial polarizations can be represented on a high-order Poincare sphere, with scalar R, L polarized vortex beams at the poles and pure vector beams around the equator [29]. Analysis of these beams requires a more detailed method beyond passing through a polarizer.

Recently, a new approach using a quantum tool was applied to measure the vector purity of a vector vortex beam [30,31,32]. In a similar manner, we measured the vector purity of our vector vortex beam based on the setup in [32] using the S-wave plate (with topological charge q = 1/2), a quarter-wave plate (only used for circular polarizations), a half-wave plate and a second SLM (with encoded hologram, Hamamatsu X13138-5785). The experimental setup is shown in Figure 2a. On-axis intensity was recorded using a CCD camera (Spiricon SP620U) by rotating the half-wave plate angle (θ_1_) and encoded hologram angle (θ_2_). Figure 2b shows the experimental normalized intensity measurement for six polarization states and six orbital angular momentum (OAM) states. The vector purity (vector quality) of a beam can be determined by entropy of entanglement (E) or concurrence (C) [30], where the concurrence C can be calculated using
(1)$$Re\left(C\right)=Re\left(\sqrt{1-s^{2}}\right)$$
where *Re* means the real part and s is the length of the Bloch vector, which is defined as [30]
(2)$$s={\left(\sum_{i}\sigma_{i}{}^{2}\right)}^{1/2}$$

Here i = 1,2,3 and $\sigma_{i}$ are the expectation values of the Pauli operators, representing a set of normalized intensity measurements. The Pauli operators $\sigma_{i}$ are computed from I and can be derived from [30]:(3)$$\sigma_{1}=\left(I_{13}+I_{23}\right)-\left(I_{15}+I_{25}\right)$$
(4)$$\sigma_{1}=\left(I_{14}+I_{24}\right)-\left(I_{16}+I_{26}\right)$$
(5)$$\sigma_{1}=\left(I_{11}+I_{21}\right)-\left(I_{12}+I_{22}\right)$$
where $I_{ij}\;$is the intensity measured on the CCD camera with associated polarization and OAM basis state. For the on-axis intensities measured on a CCD camera, the Re is superfluous; thus, Equation (1) can be simplified to C =$\;\sqrt{1-s^{2}}$. Using Equation (1), we found the concurrence (C) for our single vector vortex beam to be C = 0.95 ± 0.02 (1σ), close to an ideal vector vortex beam, C = 1. Hence the VV beam after the SWP is quite pure.

By placing the quarter-wave plate (QWP) ahead of the S-wave plate with fast axis angle α = ±45°, the output beams from the S-wave plate are left- and right- handed circularly polarized scalar vortex beams. The beam mode from vector to scalar can be obtained by rotating the QWP fast axis angle from 0° to 45°. Therefore, we measured the concurrence from a single scalar vortex beam with l = 1, C = 0.14 ± 0.01, to vector beam C = 0.95 ± 0.02. The results are shown in Figure 3.

Figure 3 shows that the experimental results do diverge from the theoretical curve expected with an incident perfect Gaussian beam, whereas for our beam, M_0_^2^ = 1.3. The fact that C ≠ 0 for a scalar vortex beam while C = 0.95 instead of unity is a reflection of the limitations of initial laser beam quality and the aberrations introduced by optical surfaces, alignment errors and SLM flatness issues. For example, it was not sensible to add the correction file to SLM2 for the data of Figure 3, which are designed for a phase-only correction, and hence a possible source of error.

## 3. Results and Discussion

### 3.1. Ablation Test on Ti-6Al-4V and Cr Thin Film with Gaussian and CV Beam

The ablation threshold fluence for a Gaussian beam can be obtained from the equation [33]
(6)$$D^{2}=2\omega_{0}^{2}\ln\left(F_{0}/F_{th}\right)$$
where *D* is the ablated crater diameter, $\omega_{0}$ is the beam waist radius, $F_{0\;}$is the peak fluence and $F_{\mathrm{th}}$ is the threshold fluence of the material. This threshold is also pulse number dependent due to incubation [33]; when focused by the flat field f-theta lens (f = 100 mm), the measured Gaussian beam diameter was$\;2\omega_{0}=22.0+0.2\;\mu m$.

#### 3.1.1. Intensity Comparison of Gaussian and CV Beams

The intensity of Gaussian and CV beams have been modelled and compared with calculations using MATLAB. The equation used to describe the ring beam amplitude is given by [34]
(7)$$E\left(r,\theta ,0\right)=E_{0}\frac{1}{\sqrt{l!}}{\left(\sqrt{2}\frac{r}{\omega_{0}}\right)}^{l}\exp\left(-\frac{r^{2}}{\omega_{0}^{2}}\right)\exp\left(il\theta \right)$$
where $E_{0}$ and $\omega_{0}$ refer to the amplitude of the normalized electric field and Gaussian beam waist, respectively; *l* is the topological charge; and θ is the phase angle, ranging from 0 to 2π. By setting l = 0, one obtains the Gaussian amplitude distribution, while setting l =1, the ring mode amplitude for CV beam profiles is obtained. The intensity distribution for l =1 is given by I = E^*^
∗ E, which immediately yields
(8)$$I\left(r\right)=E^{*}*E\;=2I_{0}\left(\frac{r^{2}}{\omega_{0}^{2}}\right)\exp\left(-2\left(\frac{r^{2}}{\omega_{0}^{2}}\right)\right)$$
where *I*_0_ is the peak intensity of the Gaussian from which the CV beam is transformed. This function goes to zero at *r* = 0, as expected. To find the ring maxima, we differentiate Equation (4), and setting the derivative to zero, we find the simple solution
(9)$$r\;=\frac{\omega_{0}}{\sqrt{2}}$$
for the position of the ring maxima, where ω_0_ is the 1/e^2^ Gaussian beam waist. Setting this value of r into Equation (8), we arrive at the absolute peak intensity of the ring beam, given by I ($r\;=\frac{\omega_{0}}{\sqrt{2}}$) = I_0_/e, where e = 2.718 is the natural logarithm. By integrating Equation (8) round 2π to obtain the total pulse power P, we obtain
(10)$$P(\mathrm{ring})\;=\;4\pi I_{0}\int r\left(\frac{r}{\omega_{0}^{2}}\right)\exp\left(-2{\left(\frac{r}{\omega_{0}}\right)}^{2}\right)\mathrm{dr}\;=\;\frac{I_{0}{\left({\pi \omega }_{0}\right)}^{2}}{2}$$

Similarly, P(Gauss) = 2π$I_{0}\int r\exp\left(-2{\left(\frac{r}{\omega_{0}}\right)}^{2}\right)\mathrm{dr}$ = $\frac{I_{0}{\left({\pi \omega }_{0}\right)}^{2}}{2}$, confirming that these beams have the same power or pulse energy.

Figure 4 shows the shows the simulated intensity distributions of the Gaussian and ring CV beam (l = 1) in MATLAB. The simulated ratio of peak intensities is$\;\;\frac{Intensity_{PG}}{Intensity_{PCV}}$ = 2.7(~e), in agreement with what is expected. Hence, if the ablation threshold is unaltered by the ring intensity distribution, the pulse energy of the CV beam should be 2.7 times higher than the Gaussian beam for the same effect. The simulations in MATLAB will be used for onward comparison with experimental results.

#### 3.1.2. Single-Beam Ablation Threshold with Gaussian and Ring Modes

One might expect the ablation threshold to be independent of spatial mode and incident polarization near normal incidence. However, this was not found to be the case recently with femtosecond ablation on dielectrics with ring mode beams [34]. This potential variation of threshold was therefore worth checking with picosecond exposure. With the SWP removed, a scalar vortex beam with l = 1 should have the same intensity distribution as a radially or azimuthally polarized beam. A single spot above the ablation threshold on thin-film chromium on glass while scanning at high speed with a vortex beam and radially polarized beam is shown in Figure 5 with optical images superimposed. For these states, the phase correction file was added to the SLM, which is not perfectly flat. The pulse energy was 22 µJ/pulse at 10 kHz repetition rate and scan speed of 400 mm/s. The similarity of the ablated spots was observed in both states. These have similar dimensions, as expected, although some slight distortion remains.

Figure 6 shows the relevant plots for obtaining the threshold energies (hence fluences) required for single-pulse modification on both titanium alloy and chromium thin film. With the Gaussian beam, D^2^ versus ln(E) yields the gradient m = 2($\omega_{0}$)^2^, allowing the peak fluence to be calculated. However, for the ring mode, at fluence threshold, a thin ring with diameter D_0_ = $\sqrt{2}\omega_{0}$ would be ablated, while above this threshold, the ring width will expand with increasing fluence. Hence, the appropriate plot for the ring beam is D^2^–D_0_^2^ = D^2^ − 2ω_0_^2^ versus ln(E), yielding the threshold ablation energy for the ring mode (l = 1) at the cut-off.

From Figure 6, the threshold ablation energy for Ti-6Al-4V with a Gaussian beam mode and CV beam mode is 0.63 ± 0.1 µJ and 1.77 ± 0.1 µJ, respectively. Hence, the ratio of the cut-off energy is 1.77 μJ/0.63 μJ = 2.8 ± 0.1, close to the value e = 2.72 calculated above, required to bring peak intensity in the ring and corresponding peak fluence to that of the Gaussian peak. The single-pulse threshold fluence of the Gaussian and CV beams from the above calculation is F_th_ = 0.32 ± 0.02 J/cm^2^ and 0.31 ±0.02 J/cm^2^, respectively—in excellent agreement. As the cut-off energy for a Cr-coated thin film with a Gaussian beam mode and CV beam mode is 2.67 ± 0.2 µJ and 6.22 ± 0.2 µJ, respectively, the ratio of the cut-off energy is 6.22 μJ/2.67 μJ = 2.3 ± 0.2, lower than expected, but this substrate may be more sensitive to CV beam azimuthal intensity non-uniformity. The corresponding ablation thresholds of the Cr thin film with Gaussian and CV single pulse are 1.38 ± 0.02 J/cm^2^ and 1.73 ± 0.02 J/cm^2^, respectively—not in agreement within the quoted uncertainties—so that in this case, these have been underestimated. The best focal plane for micro-structuring on the thin film with ring modes is fairly close to that of the Gaussian since we are dealing essentially with collimated beams. The best plane was determined by optimizing the substrate height to yield the roundest ablation spots, and the remaining aberrations cause non-uniform cylindrical intensity and ablation, observed on the sensitive thin film. This may explain why the ablation threshold with Gaussian and vortex beams do not agree within the experimental error.

We modelled ring beams with higher l values and the resulting intensity distributions. The details are shown in Table 1. These functions show that peak ring intensities drop with increasing topological charge l—while the peak radii increase. Integrating these functions round 2π, all yield the same power (P = I_0_πω_0_^2^/2) or pulse energy. Again, ω_0_ is the Gaussian beam 1/e^2^ radius, from which these intensity profiles are generated, and I_0_ the Gaussian peak intensity.

Figure 7 shows the ablation plots of D^2^–D_0_^2^ vs. ln(E) for l = 1 and l = 2 vortex beam.

The threshold ablation energy for Ti-6Al-4V for vortex beams with l = 1 and l = 2 was 1.77 ± 0.1 µJ and 2.41 ± 0.1 µJ, respectively; hence the ratio of the cut-off energy is 2.41 μJ/1.77 μJ = 1.36 ± 0.1. This is close to the value e/2 ≈ 1.35, which can be calculated from Table 1. Hence the ablation threshold of titanium alloy for l = 2 is similar to that for l = 1. Relative to the Gaussian beam, the ratio of the cut-off energy l = 1 is 1.77 μJ/0.63 μJ = 2.81 ± 0.1, slightly higher than expected (e ≈ 2.7), while for l = 2, the ratio is 2.41 μJ/0.63 μJ =3.81 ± 0.1, close to that expected (e^2^/2 ≈ 3.7) from Table 1.

### 3.2. LIPSS Formation on Titanium Alloy with CV Beams

Figure 8 shows optical images of LIPSS formation with linearly polarised Gaussian and vortex beams with l = 1 and l = 3. The ablated widths were 20 μm, 45 μm and 82 μm with scan speeds s = 20 mm/s, 33 mm/s and 36 mm/s, respectively. Processing rates or area coverage were therefore increased by a factor of 3.7 (l = 1) and 7.4 (l = 3), respectively.

### 3.3. Multiple Cylindrical Vector (CV) Beam Generation

Two binary gratings on the SLM, 1 × 5 and 2 × 2 Dammann gratings, were generated by MATLAB and were used to create uniform multiple cylindrical vector beams with the S-wave plate. The transition points within a single period as described in [28] were used for the 1 × 5 Dammann grating and 2 × 2 Dammann grating generation. Figure 1c shows the details of the generated Dammann gratings. A Dammann grating consists of many grating unit cells, each having two grey levels (normally white and black, corresponding to 0 and π). The intensity and uniformity of the output beam arrays based on a Dammann grating is affected by the grey level values of the grating unit cell. The uniformity of the parallel spots generated can be measured with the equation η =$\;\frac{\sigma_{i}}{a_{i}}$, where $\sigma_{i}$ and $a_{i}$ are the standard deviation and the arithmetic mean of the ablated spot diameters, respectively. The uniformity of the spots reached the highest at the grey level of 107 (for the Hamamatsu 10468-03, the full 2π phase modulation corresponds to grey level 212), which was derived from the machined results. Thus, the grey level value of 107, which corresponds closely to π, was selected for both Dammann gratings used in this study to achieve the highest uniformity. The measured diffraction efficiency of the calculated 1 × 5 Dammann grating and 2 × 2 Dammann grating with a grey level value of 107 is 86% and 84%, respectively. Multiple cylindrical vector beams can then be generated by combining the optimized binary Dammann grating on the SLM with the S-wave plate. However, the 0th order is still present at the centre of the beam array. The residual zero order of a diffractive pattern can degrade the uniformity of a parallel machined pattern as it will typically have higher power compared with that of the other diffractive orders. We eliminated the zero order by adding a CGH, Fresnel lens (FZL), to effectively place the 0th order well away from the processing plane, bringing its peak fluence below the ablation threshold [27].

### 3.4. Parallel Processing on a Ti-6Al-4V Surface Using Uniform Multiple Cylindrical Vector Beam Arrays

As shown in Figure 9, multiple laser beam spots (1 × 5 and 2 × 2 arrays) were machined on polished Ti-6Al-4V with radial and azimuthal polarizations. Here, laser repetition rate was set at 10 kHz with a 40 µJ/pulse and 45 pulses per spot. The images show the measured intensity profiles in (a), analysed through a polarizer (b), while the plasmonic LIPSS are shown in the corresponding SEM images (c) and (d).

The above LIPSS generation confirms the polarization states when one assumes that these low-frequency LIPPS form at right angles to the local electric vectors. The multiple CV spots produced by this method compare reasonably well to other SLM-based methods of parallel CV spot generation [21]. However, perfect LIPSS structures were not achieved in this work. This is likely due to the beam mode issue, some residual beam distortion and setup alignment error.

### 3.5. Similarity of Each Spot: Cosine Similarity

Cosine similarity is a method to measure the similarity between two non-zero vectors of a product or image. If the cosine angle of two non-zero vectors of 0° for an image is 1, it would mean that they have the same magnitude and direction, so the compared images are the same. To compare the similarity of the machined diffractive spots, a MATLAB code based on the cosine similarity method was used. The similarity can be defined as
(11)$$\mathrm{Similarity}=\cos(\theta )=\frac{\vec{A}\;\cdot \;\vec{B}}{\left||\vec{A}||\;||\vec{B}|\right|}=\frac{\sum_{i=1}^{n}A_{i}B_{i}}{\sqrt{\sum_{i=1}^{n}A_{i}^{2}}\sqrt{\sum_{i=1}^{n}B_{i}^{2}}}$$
where *A* and *B* are two vectors of attributes of the image, and $A_{i}$ and $B_{i}$ are components of vectors *A* and *B*, respectively. As the SEM images are grey level, a MATLAB code was used to generate a grey-scale histogram of the images. These grey-scale histograms can be compared using MATLAB to create a cosine similarity.

To decrease the measurement error, each machined spot image was taken with the same magnification factor to ensure that the details were clearly presented. The first spot of each kind of polarization was selected as a reference in the 1 × 5 and 2 × 2 spot arrays. Then, the other spots were compared to the reference spot. For example, the results of comparison of the reference spot with the second spot would be group 1, the results of the reference spot with the third spot would be group 2, and so on. Therefore, 10 result groups can be achieved in the comparison of 1 × 5 spot arrays, and 6 result groups can be achieved in 2 × 2 spot arrays. The details of the similarity results for the two spots are provided in Figure 10, showing a high degree of similarity in the array spots micro-machined with azimuthal polarization.

A summary of the comparison of similarities is shown in Figure 11a,b, with very encouraging results. Using the ablated patterns to estimate similarities and hence indicating the uniformity of arrays and their corresponding intensity distributions is a novel approach, complementing polarization analysis.

For high similarity, the cosine angle should be close to 1 and the values should have a small spread. Based on the histogram data in Figure 11, the similarity of each spot in the 1 × 5 cylindrical vector beam arrays and the 2 × 2 cylindrical vector beam arrays are high, with cosine similarity 0.9 ≤ cosθ ≤ 0.995. As the cosine angle is >0.9, the compared spots can be regarded as almost the same. By comparing with the machined spot size of polished material surface, the beam arrays produced were found to have a good similarity with spots within 10% of each other in terms of structure.

The azimuthal polarization in the 1 × 5 array has cosθ > 0.975, while radial is consistently lower, >0.96. On the other hand, this trend is reversed with the 2 × 2 array, where radial similarity demonstrates cosθ > 0.98, while azimuthal can drop to cosθ = 0.9.

## 4. Laser Surface Texturing

### Continuous Scanning with Different Polarization States

The setup was configured to scan a series of line patterns, where polarization was sequentially changed per line during scanning. Figure 12 shows optical micrographs of the surfaces produced and confirms the polarization states with LIPSS orientations. The SEM microscopic analysis confirmed the expected LIPSS orientations.

The depth of the shark skin surface features, resulting from overscanning, were approximately 100–150 nm deep.

## 5. Conclusions

The generation of uniform multiple radial or azimuthal polarized beam arrays with 1 × 5 and 2 × 2 ring beams was demonstrated by combining a nanostructured geometric phase waveplate with a spatial light modulator addressed with Dammann array gratings. Uniformity was improved by adding an FZP to defocus the zero order. Using a quantum toolkit, the vector purity (concurrence and entropy) of the vector vortex beam was found to be C = 0.95 ± 0.02. Ablation threshold measurements typically assume a Gaussian beam profile. This method was modified to accommodate ablation with ring modes, which demonstrated that when absolute intensity (fluence) is matched, the single-pulse ablation thresholds are essentially unaltered. For example, the measured ablation threshold of Ti-6Al-4V with Gaussian and CV single pulse was 0.32 ± 0.02 J/cm^2^ and 0.31 ± 0.02 J/cm^2^, respectively, while the ratio of pulse energies was found to be 2.8, close to the theoretically expected value of e = 2.72. In the case of chromium thin film on glass, the ratio was 2.3, lower than expected by 15%. This difference likely arises from the sensitivity of the thin-film substrate to CV beam azimuthal intensity non-uniformity. The Cr thin-film single-pulse ablation thresholds with Gaussian and CV single pulse were 1.38 ± 0.02 J/cm^2^ and 1.73 ± 0.02 J/cm^2^, respectively, not in agreement within the uncertainties, so that in this case, these were underestimated. The vortex beam divergence is higher than Gaussian, with a beam quality factor M^2^ = M_0_^2^ + mod l [34], so that with l  = 3, M^2^ = 1.3 + 3 = 4.3. The Rayleigh length here for the Gaussian beam (with radius 0.4 cm), Z_R_ =$\;\frac{\;\pi \omega_{0}^{2}}{\lambda }$/M_0_^2^ ~ 36.5 m, while with l  = 3, M^2^ = 4.3 and corresponding Z_R_ (l = 3) ~11.0 m, still very significant. The resulting full-angle beam divergence θ_FA_ = (2ω_0_/Z_R_) = 0.22 mR (Gaussian) and 0.72 mR (l = 3), respectively. Over the beam paths here of a few metres (and with the use of 4f optics), this expansion was not an issue.

The expected ring intensity distribution (radial/azimuthal polarization) created from a Gaussian was modelled, and it indicated that the ring peak intensity reaches I_0_/e, where I_0_ is the Gaussian peak intensity and e is the natural logarithm. The ring maximum also occurs at r = $\frac{\omega_{0}}{\sqrt{2}}$. The threshold ablation energy for Ti-6Al-4V for vortex beams with l = 1 and l = 2 was 1.77 µJ and 2.41 µJ, respectively; hence the ratio of the cut-off energy is 2.41μJ/1.77μJ =1.36 ± 0.1. This is close, allowing for experimental error, to the value e/2 ≈1.35, expected theoretically; hence the ablation thresholds are similar to those for the Gaussian beam. Relative to the Gaussian beam, the ratio of the cut-off energy for comparing l = 1 is 1.77 μJ/0.63 μJ = 2.81 ± 0.1, slightly higher than expected (e ≈ 2.7), while for l = 2, the ratio is 2.41 μJ/0.63 μJ = 3.82 ± 0.1, close to that expected (e^2^/2 ≈ 3.7). Hence, the ablation thresholds of Ti-6Al-4V with picosecond pulse length ring beams of different *l* are very similar. Physically, this is to be expected since peak intensity and fluence (along with material physical properties) determine the ablation threshold. If the ablation threshold were dependent on intensity distribution, a change in the light–matter coupling would be inferred. This appears unlikely. However, recent results with femtosecond laser pulses on dielectrics give ablation thresholds depending on beam mode [35]. With ultrafast pulses, heat diffusion is minimized along with melting, so that ablation becomes a deterministic process dependent on the peak intensity (fluence) and material physical properties. Ashkenasi et al. demonstrated the equivalence of ablation thresholds independent of Gaussian beam diameter [36].

The method of cosine similarity was used to analyse the comparison of each spot. The beam arrays produced were found to have a very high similarity with spots within 10% of each other in terms of structure. This indicates that excellent repeatability was achieved. The machined footprints have the desired polarization state as confirmed by LIPSS. Complex shark skin-like LIPSS appear with specific laser parameters during beam scanning due to the overlap of vector fields. The complex LIPSS generated by overlaid polarization states are a composite topography, rather than being wiped and reset by each other, and 5 × 5 or even more ring mode spots can be machined if enough laser energy is applied. Recently, multiple parallel spots have been employed, with a cooled SLM, operating at a high average power at the 100 W level and demonstrating multi-beam thin-film patterning at 80 cm^2^/s [37]. Our approach, presented in this work, has potential application in multiple ring mode laser processing with an SLM under high average power.

Scanning polished titanium samples (Ti-6Al-4V) with ring modes with identical peak intensities of a Gaussian accelerates LIPPS surface processing significantly, so that beam shaping, using higher available powers, can increase surface processing speed. The method used in this research has potential applications in large-area LIPSS generation for laser security marking, material wettability modification as well as multiple micro-drilling.

## Figures and Tables

**Figure 1 micromachines-12-00376-f001:**
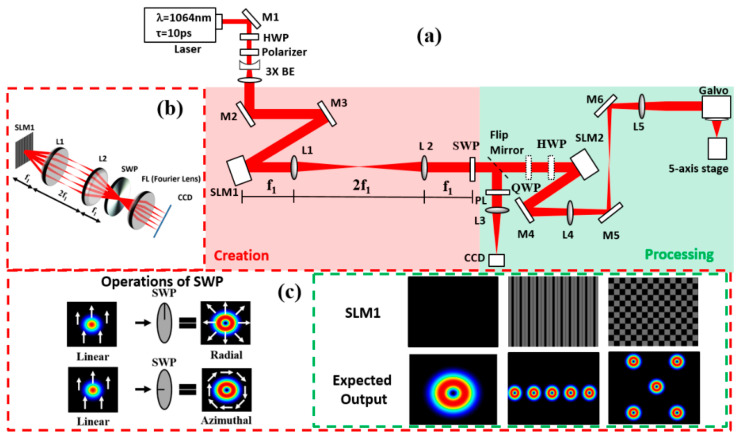
(**a**) Experimental setup with an S-wave plate. (**b**) Experimental concept. (**c**) Operations of the S-wave plate and the expected output with the addressed computer-generated hologram (CGH) on the spatial light modulator (SLM). A polarizer (PL) could be inserted ahead of lens L3 and CCD camera for polarization analysis.

**Figure 2 micromachines-12-00376-f002:**
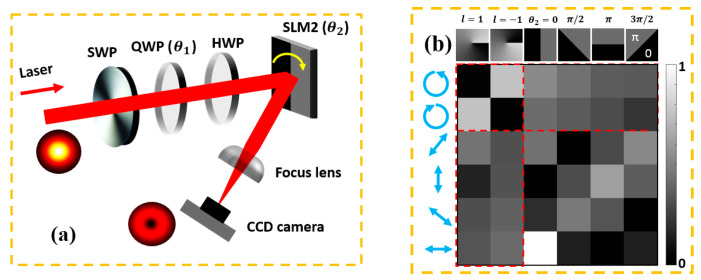
(**a**) Optical setup for vector purity measurement. SWP, S-wave plate; QWP, quarter-wave plate; and HWP, half-wave plate. The focal length for the focus lens is 500 mm (20 μJ/pulse, 10 kHz). (**b**) Normalized experimental measured intensities with a combination of a quarter-wave plate and half-wave plate. The concurrence of the vector beam was calculated by the measurements in the outlined red dash lines, OAM projections (horizontal) or polarization projections (vertical).

**Figure 3 micromachines-12-00376-f003:**
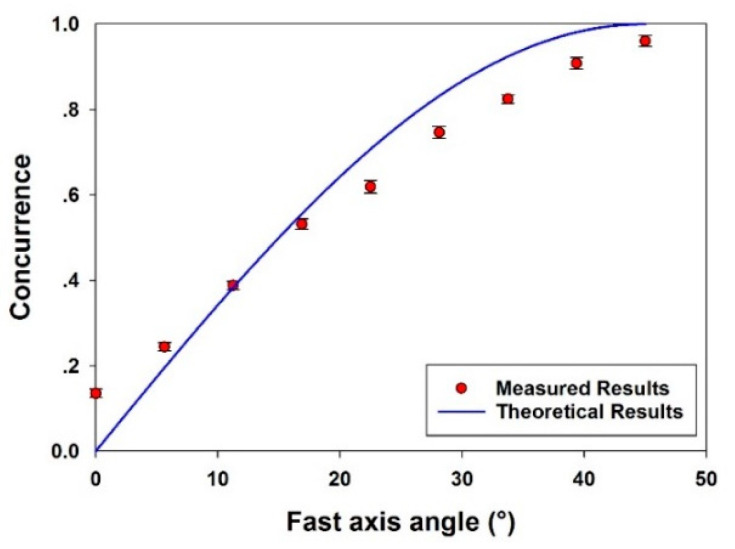
Measured concurrence of a single beam from scalar mode (C = 0.14) to vector mode (C = 0.95). Fitted with a pure cos^2^θ function. The difference between theory and experiment does indicate some residual aberration on the beam (20 μJ/pulse, 10 kHz).

**Figure 4 micromachines-12-00376-f004:**
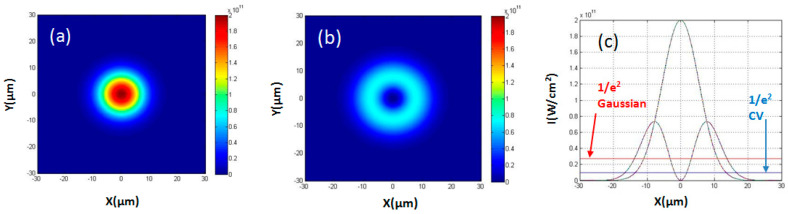
The simulated intensity distribution in MATLAB: (**a**) a Gaussian distribution, l = 0; (**b**) a cylindrical vector (CV) beam distribution, l = 1 with the same total pulse energy. (**c**) Cross-section view of the Gaussian distribution and the CV beam distribution. The two lines show the 1/e^2^ intensity levels, the red line for the Gaussian distribution beam and the blue line for the CV beam distribution.

**Figure 5 micromachines-12-00376-f005:**
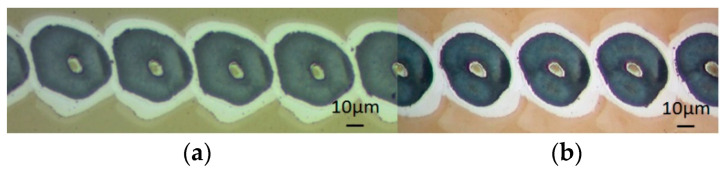
Single-pulse ablation with (**a**) a vortex beam (l = 1) and (**b**) a CV beam (radial) on a Cr-coated thin film. By overlapping the two images, the beam size for l = 1 and the CV beam is similar (30 µJ, 10 kHz, 500 mm/s).

**Figure 6 micromachines-12-00376-f006:**
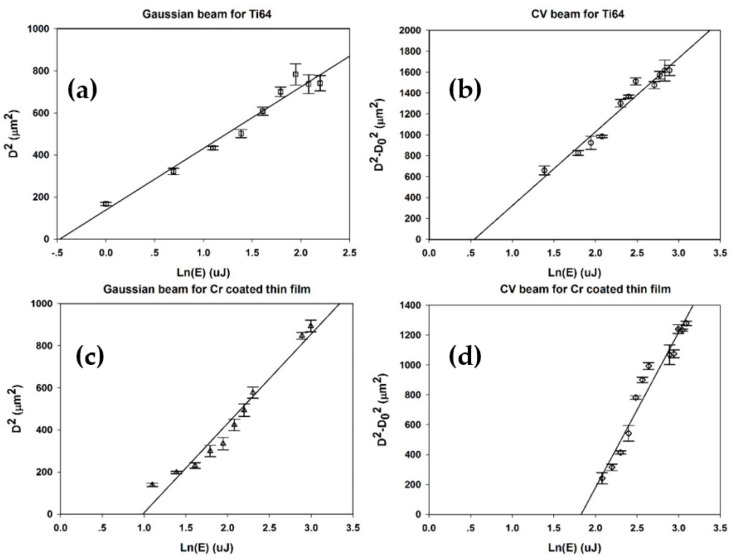
(**a**) Plot of squared diameter D^2^ versus ln(E) for Gaussian beam ablation threshold fluence calculation of Ti-6Al-4V with single pulse. (**b**) Plot of D^2^–D_0_^2^ versus ln(E) for radially polarized CV beam ablation threshold energy calculation of Ti-6Al-4V with single pulse. (**c**) Plot of squared diameter D^2^ versus ln(E) for Gaussian beam ablation threshold fluence calculation of Cr-coated thin film with single pulse. (**d**) Plot of D^2^–D_0_^2^ versus ln(E) for CV beam ablation threshold fluence calculation of Ti-6Al-4V with single pulse.

**Figure 7 micromachines-12-00376-f007:**
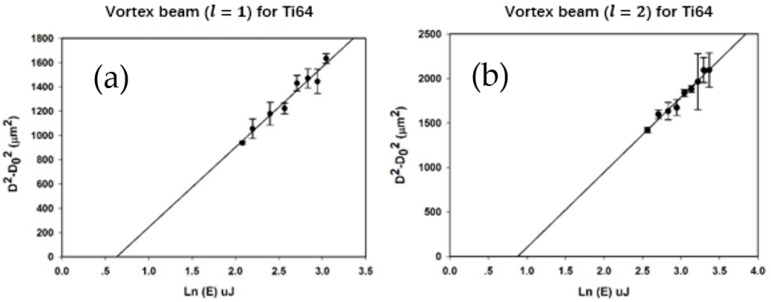
(**a**) Plot of squared diameter D^2^–D_0_^2^ (or D^2^–2ω_0_^2^) versus ln(E) for vortex beam (l = 1) ablation threshold fluence calculation of Ti-6Al-4V with single pulse. (**b**) Plot of D^2^–D_0_^2^ (or D^2^−4ω_0_^2^) versus ln(E) for vortex beam (l = 2) ablation threshold energy calculation of Ti-6Al-4V with single pulse.

**Figure 8 micromachines-12-00376-f008:**
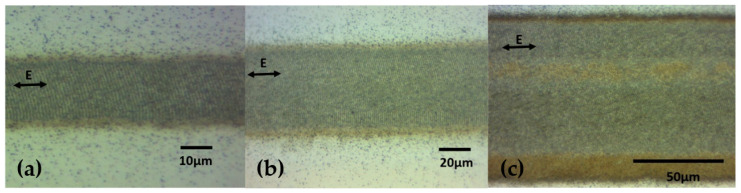
Laser-ablated polished Ti-6Al-4V with (**a**) Gaussian beam, 15 mW, (**b**) l = 1 vortex beam, 42 mW, and (**c**) l = 3 vortex beam, 70 mW. All beams have the same peak intensities (fluence).

**Figure 9 micromachines-12-00376-f009:**
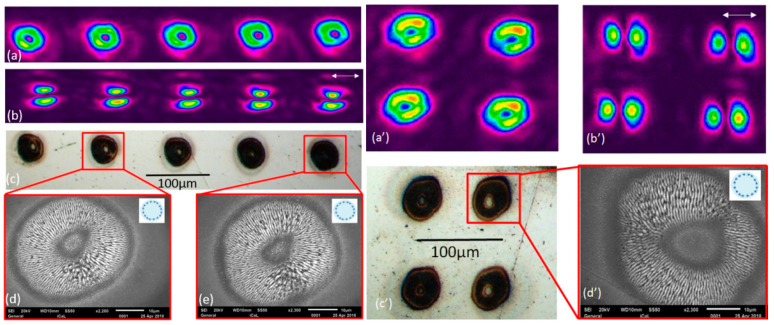
Azimuthally polarized uniform 1 × 5 vector beam array: (**a**) beam profiler image; (**b**) beam profiler image with a polarizer analyser. The white arrow represents the direction of the fast axis of the polarizer analyser. (**c**) Machined uniform 1 × 5 vector beam arrays on a Ti-6Al-4V sample. The diameter of each spot is approximately 40 µm. (**d**,**e**) The SEM images (10 kHz, 40 µJ/pulse, 45 pulses). Azimuthally polarized uniform 2 × 2 vector beam array: (**a’**) beam profiler image; (**b’**) beam profiler image with a polarizer analyser. The white arrow represents the direction of the fast axis of the polarizer analyser. (**c’**) Machined uniform 1 × 5 vector beam arrays on a Ti-6Al-4V sample. The diameter of each spot is approximately 40 µm. (**d’**) The SEM images (10 kHz, 40 µJ/pulse, 45 pulses). The scale bar in the SEM images is 10 μm.

**Figure 10 micromachines-12-00376-f010:**
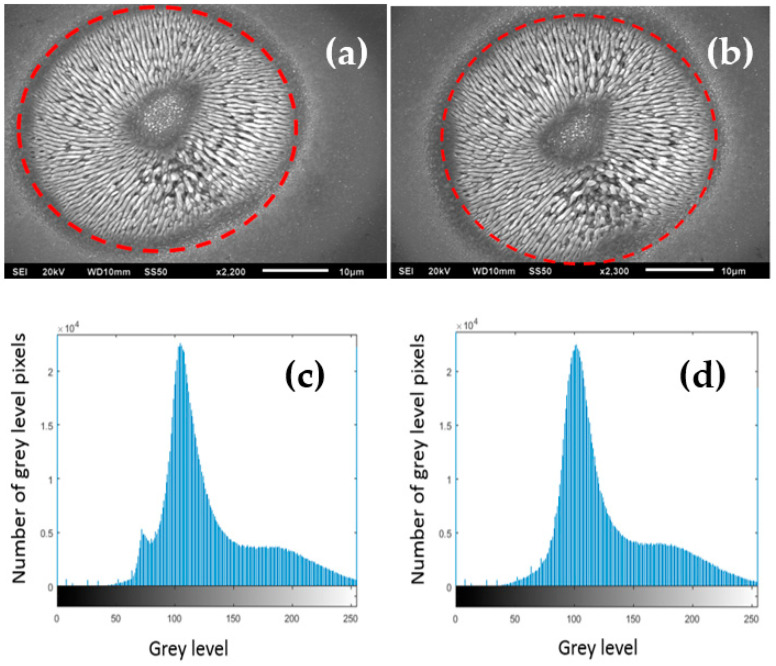
Examples of grey-scale histograms that can be extracted from the SEM images for azimuthal polarization. (**a**) Measured area (inside of the red dash line). (**c**) The grey-scale histogram extracted from (**a**). (**d**) The grey-scale histogram extracted from (**b**).

**Figure 11 micromachines-12-00376-f011:**
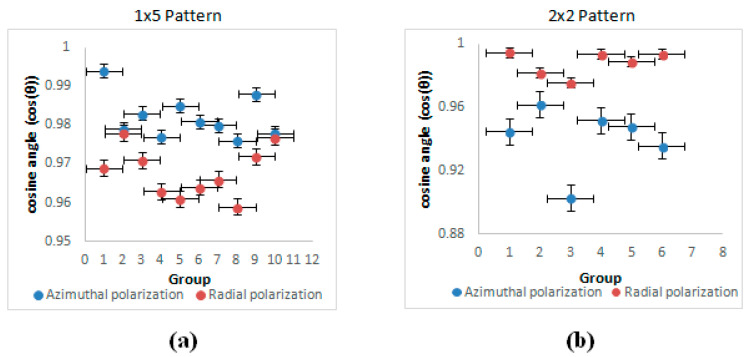
(**a**) Line graph of cosine similarity for 1 × 5 cylindrical vector beams. (**b**) Line graph of cosine similarity for 2 × 2 cylindrical vector beams.

**Figure 12 micromachines-12-00376-f012:**
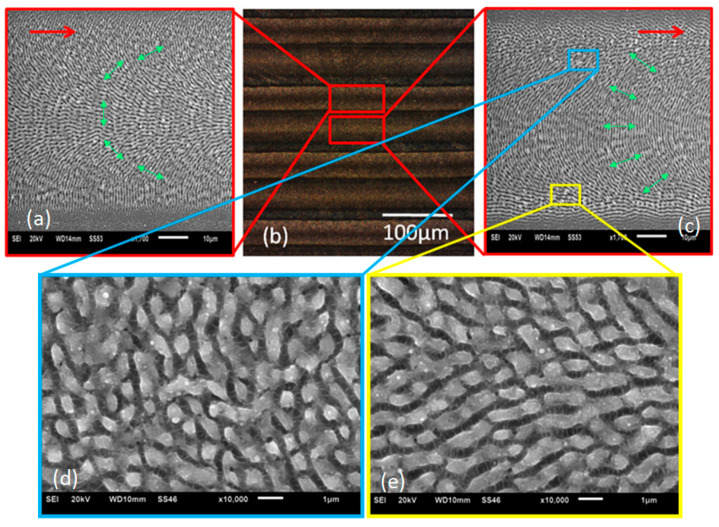
Continuous scanning results with different polarization states. (**a**) SEM result of azimuthal polarization (10 kHz, 40 µJ/pulse, 10 mm/s). (**b**) Optical micrograph of line patterns with side illumination. (**c**) SEM result of radial polarization (10 kHz, 40 µJ/pulse, 10 mm/s). The red arrows represent the laser scanning direction. The green arrows represent the E field. (**d**,**e**) Images at higher magnifications of areas inside the blue and red squares, showing shark skin-like morphologies.

**Table 1 micromachines-12-00376-t001:** Calculated intensity distributions with different topological charges m and corresponding radius *r*, where the ring intensity is maximum. Here, $\omega_{0}$ and $I_{0}$ is the beam waist radius and the peak intensity of the Gaussian mode, respectively.

Topological Charge l	*r* (I_peak_)	Intensity Expression	Peak Intensity
1	$\frac{\omega_{0}}{\sqrt{2}}$	$2I_{peak}*\frac{r^{2}}{\omega_{0}^{2}}*\exp\left(-2\frac{r^{2}}{\omega_{0}^{2}}\right)$	$\frac{I_{0}}{e}$
2	$\omega_{0}$	$2I_{peak}*\frac{r^{4}}{\omega_{0}^{4}}*\exp\left(-2\frac{r^{2}}{\omega_{0}^{2}}\right)$	$\frac{2I_{0}}{e^{2}}$
3	$\frac{\sqrt{3}\omega_{0}}{\sqrt{2}}$	$\frac{4}{3}I_{peak}*\frac{r^{6}}{\omega_{0}^{6}}*\exp\left(-2\frac{r^{2}}{\omega_{0}^{2}}\right)$	$\frac{9I_{0}}{2e^{3}}$
4	$\frac{\sqrt{4}\omega_{0}}{\sqrt{2}}$	$\frac{2}{3}I_{peak}*\frac{r^{8}}{\omega_{0}^{8}}*\exp\left(-2\frac{r^{2}}{\omega_{0}^{2}}\right)$	$\frac{32I_{0}}{3e^{4}}$

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
