# Peer review of "Laser–Material Interactions of High-Quality Ultrashort Pulsed Vector Vortex Beams"

_micromachines, 2021, doi:10.3390/mi12040376_

Round 1

Reviewer 1 Report

In this work, diffractive multi-beams based on 1x5 and 2x2 binary Dammann gratings applied to a Spatial Light Modulator (SLM) combined with a nanostructured S-waveplate have been used to generate uniform multiple cylindrical vector beams with Radial and Azimuthal polarizations. The vector quality factor (concurrence) of the single vector vortex beam was found to be C= 0.95 ± 0.02 hence showing a high degree of vector purity. The multi-beams have been used to ablate polished metal samples (Ti-6Al-4V) with laser induced periodic surface structures (LIPSS) which confirm the polarization states unambiguously. This work is interesting and the results are sound. I suggest it can be accepted after the following questions are properly solved.

  1. What are the advantages of cylindrical vector beams and ring mode vortex beams on laser ablation compared with Gaussian beam?
  2. As we know, the divergence of vortex beam is faster than Gaussian beam. Therefore, it is suggested that the authors give some discussions about the working distance of vortex beam in laser ablation. How to further improve the working distance of vortex beam?
  3. Some relevant references may help enrich the introduction, e.g., Opto-Electronic Advances 4, 200072 (2021), Opto-Electron Adv 3, 190037 (2020), Optica 7(5), 518-526 (2020). Materials, 2021, 14(4): 1022.

Reviewer 2 Report

The authors report ps laser ablation of materials to generate complex  structuration of its surface. The experiment involves LC SLM and diffractive optics for shaping the polarization and phase of the incident vector vortex beams. The  single and multiple array beam shaping function is of interest for industrial applications in high speed surface material processing. The results of experiments are correctly introduced and discussed in the paper, but several questions merit to be considered with attention, as follows : 

- Give the energy per pulse of the laser, beam quality ? 

- It is noted the use of a Nanostructured SWP component. Include details about the operating mode of the SWP and the structure of the device. It is not clear which is the required distance between SWP and SLM2 - SWP diffracts on the SLM2 in Fig 2 ?        

- The vector quality or vector purity parameter is introduced in the text. It is required to develop the physical significance of this parameter of the vortex beam which is relevant to the given relations 1 to 5 and thus leading to the direct conclusion that VV beam is quite pure. Sensitivity of the VV beam to the defocus on the thin film? How aberrations of the optical components affect the quality and profile of the VV beam on the coated thin film. Consequence of aberrations on the energy threshold ? Note Fig 2b and not Fig3b                             

- Which is the measured efficiency of the Dammann grating displayed on the SLM. It is not clear that the FZL on the SLM only defocus the zero order and not the mutiple orders ? Which is the average value of the depth of the shark lines in Fig 12.  

To conclude the manuscript includes useful datas and novel experimental conditions to optimize the pulse energy, the vortex beam profile as well as the 2D parallel operation on the film surface. It brings significant results to contribute to laser surface material processing applications. As noted in the review it is required that the authors take account of the comments and questions in the revised  form of the paper before publication in the journal.  

Round 2

Reviewer 2 Report

The authors have clearly answered to questions and remarks. I note a contribution to precise several points which confirm interest of VV for laser material structuration. The paper is now well suited for publication in the journal.